# Cardiovascular Outcomes of ST-Elevation Myocardial Infarction (STEMI) Patients without Standard Modifiable Risk Factors (SMuRF-Less): The Intermountain Healthcare Experience

**DOI:** 10.3390/jcm12010075

**Published:** 2022-12-22

**Authors:** Jeffrey L. Anderson, Stacey Knight, Heidi T. May, Viet T. Le, Jawad Almajed, Tami L. Bair, Kirk U. Knowlton, Joseph B. Muhlestein

**Affiliations:** 1Intermountain Medical Center Heart Institute, Murray, UT 84107, USA; 2Department of Internal Medicine, University of Utah School of Medicine, Salt Lake City, UT 84132, USA; 3Rocky Mountain University of Health Professions, Provo, UT 84606, USA

**Keywords:** myocardial infarction, ST-elevation, risk factors, STEMI

## Abstract

Studies primarily outside the United States have reported that SMuRF-less STEMI patients are surprisingly common (14–27%) and have a worse in-hospital/short-term prognosis. Given potential demographic and management differences over time and in the US, we aimed to identify the proportion and outcomes of SMuRF-less STEMI patients in a large US healthcare population. Patients with a first STEMI presenting to Intermountain Healthcare catheterization laboratories between 2001–2021 were included. SMuRF included a clinical diagnosis of, or treatment for, hypertension, hyperlipidemia, diabetes, and smoking. Follow-up MACE were defined as death, MI, and heart failure hospitalization (HFH) by 60 days and long-term. Qualifying STEMI patients totaled 3510, 26.2% (919) with no SMuRF. SMuRF-less patients were younger, more frequently male, and had fewer comorbidities. Neither total MACE (adj HR 0.95, *p* = 0.72) nor death (adj HR 1.06, *p* = 0.69) differed by SMuRF status at 60 days. Long-term outcomes were more frequent in SMuRF patients, which remained significant for total MACE (adj HR 0.83, *p* = 0.02) and HFH (HR 0.36, *p* = 0.0005) after adjustment for baseline differences other than SMuRF. Results were consistent through subgroup and sensitivity analyses. In this moderately large US healthcare population, SMuRF-less STEMI presentation was confirmed to be common (26.2%). However, unlike earlier, mostly non-US reports, adjusted short-term outcomes were similar, and long-term outcomes were more favorable. Further studies to increase understanding, recognition, and treatment of risk factors in SMuRF-less subjects and to optimize STEMI management are indicated.

## 1. Introduction

The recent recognition that an important minority of patients presenting with ST-elevation myocardial infarction (STEMI) do not have standard modifiable risk factors (SMuRF) has sparked interest in defining their prevalence across various populations and assessing their acute and chronic prognosis. Initial studies outside of the United States reported that SMuRF-less STEMI patients are surprisingly common (14–27% of STEMI cases) and had a worse in-hospital/short-term prognosis [1,2]. Given potential demographic and management differences in the United States and across healthcare systems and evolving management strategies, we aimed to identify the proportion and outcomes of SMuRF-less STEMI patients within a large US healthcare population.

## 2. Materials and Methods

**Study aims and IRB approval:** Our primary study aim was to assess the prevalence of SMuRF-less patients presenting with STEMI to the emergency departments and catheterization laboratories of Intermountain Healthcare and to determine their cardiovascular prognosis compared to patients with standard modifiable risk factors (SMuRF). This retrospective observational database study was approved by the Intermountain institutional review board (IRB) with a waiver of consent.

**Healthcare system and STEMI pathway:** Intermountain Healthcare is a nonprofit, integrated healthcare system that included 24 hospitals, 215 clinics, and an affiliated health insurance company in Utah, Idaho, and Nevada during the study period. Intermountain Healthcare has an extensive and long-standing (>25 years) centralized electronic medical records system, the electronic data warehouse (EDW), and a complementary, integrated catheterization laboratory records database. Intermountain Healthcare has an integrated STEMI-care pathway, whereby STEMI patients are triaged directly to a PCI-capable facility where geographically feasible or, in a facilitated way, from a secondary facility to a PCI-capable facility.

**Study population and SMuRF definition:** Patients with a first STEMI presenting to Intermountain Healthcare catheterization laboratories between 1 June 2001 and 31 January 2021 comprised the study population. 

SMuRF was defined as a clinical diagnosis of, or treatment for, hypertension, hyperlipidemia, diabetes, and/or smoking (current or former), based on the Intermountain EDW and the catheterization laboratory records database. 

**Study endpoints:** The primary study endpoints were major adverse cardiovascular events (MACE), which included all cause death, myocardial infarction (MI), and heart failure hospitalization (HFH) within 60 days of STEMI, and long-term MACE (to end of follow-up, i.e., 3 March 2021). Secondary endpoints included a composite of these same factors and individual factors during long-term follow-up. 

**Statistical analysis:** The prevalence of SMuRF-less patients presenting with STEMI and their short-term (60-day) and long-term (to end-of-follow-up) risk of MACE were determined and compared to patients presenting with SMuRF. The chi-square statistic and t-test were used to examine differences in baseline characteristics and medications for those patients with and without SMuRF. Cox hazard regression (unadjusted and adjusted [adj]), was used for both short- and long-term MACE.

## 3. Results

**Study patients and demographics:** Qualifying STEMI patients totaled 3510, of which 919 (26.2%) were SMuRF-less. Baseline characteristics of STEMI patients by SMuRF status are presented in Table 1. SMuRF-less patients were younger, more frequently male, and were predominately White/Caucasian. SMuRF-less patients had fewer co-morbidities than SMuRF patients (i.e., atrial fibrillation, COPD, family history of CAD, with trends to less heart failure and depression (Table 1)). 

**STEMI treatment:** STEMI treatment is summarized in Table 2. PCI was performed frequently and equivalently in SMuRF-less (85.8%) and SMuRF patients (87.5%). Discharge medications included near universal prescription of aspirin (>95%) and other antiplatelets (>95%) in both groups, a high and similar percentage of beta blockers (84.0% vs. 85.6%), and ACEIs or ARBs in a large majority, slightly favoring SMuRF patients (67.1% vs. 62.7%; *p* = 0.01).

**Short-term MACE outcomes:** Table 3 summarizes cardiovascular outcomes of STEMI Patients by SMuRF status. Within 60 days, there were a total of 210 cardiovascular events, with rates of 7.73% in SMuRF-less patients and 8.10% in SMuRF patients. The adjusted hazard ratio (adj HR) for SMuRF-less patients was 0.95 (95% CI 0.72, 1.25; *p* = 0.72). All-cause death was the most frequent outcome (*n* = 170), and again, the adjusted hazard risk was similar for SMuRF-less and SMuRF patients (HR = 1.06 (0.79, 1.42); *p* = 0.69). There were numerically fewer recurrent MIs (NS) and significantly fewer heart failure hospitalizations in SMuRF-less patients (0.2% vs. 1.0%, *p* = 0.03).

**Long-term STEMI outcomes:** Long-term outcomes at a median (IQR) follow-up time of 4.8 (1.1, 9.9) years were more frequent in SMuRF patients (Table 3 and Figure 1). Total MACE (21% vs. 31%) and each individual MACE category were fewer long-term in SMuRF-less patients, including all-cause death (17.4% vs. 24.9%), MI (4.6% vs. 6.9%), and HFH (1.4% vs. 4.8%). These differences resolved for death and MI after adjustment for baseline differences other than SMuRF but not for total MACE (adj HR 0.83 (0.71, 0.98); *p* = 0.02) or for HFH (adj HR 0.36 (0.20, 0.64); *p* = 0.0005).

**Subgroup and sensitivity analyses**: Long-term MACE by SMuRF status was examined for several patient subgroups (Figure 2). For all these subgroups SMuRF-less patients were less likely to have long-term MACE. Although males and subjects <60 years old were the only two SMuRF-less subgroups showing statistically significant decreases in long-term MACE, there were no significant interactions between subgroup pairs.

We also examined outcomes by the number of SMuRF factors. Demographic and treatments did differ by the number of SMuRF factors (Appendix A). Adjustment for these baseline differences resolved differences in 60-day MACE by SMuRF status (Table 4). However, each one increase in SMuRF count was associated with an increase in the long-term MACE (adj HR per increase in risk factor = 1.16 (1.09, 1.23), *p* < 0.0001) (Table 4). The individual long-term outcomes of death and heart failure hospitalization also increased with increasing numbers of SMuRF factors (Table 4). Inclusion of former smokers in the no SMuRF category did not change outcome comparisons (Appendix A). 

## 4. Discussion

**Summary of Key Study Findings:** Our study observed 3 key findings: First, the prevalence of SMuRF-less STEMI is high—over a quarter of patients in the Intermountain experience. Second, the in-hospital/short-term (60-day) rates of MACE and death are similar to those of STEMI patients with standard modifiable risk factors. Third, long-term outcomes favor SMuRF-less patients, with lower rates of total MACE, death, MI, and HFH, although differences resolve for MACE, death, and MI, but not for HFH, after adjustment for baseline differences. The findings were consistent through multiple subgroup and sensitivity analyses.

These findings underscore the need for better primary risk prediction algorithms, so that preventive therapies can be provided for SMuRF-less subjects at STEMI risk as well as for those with traditional risk factors. However, our findings are reassuring in indicating that, in the current era of rapid triage and reperfusion therapy plus guideline-directed medical treatment of ACS, an increased short-term risk can be avoided. 

**Literature Comparisons:** In a 2017 report, Vernon et-al drew attention to an increased proportion of STEMI patients presenting without standard modifiable cardiovascular risk factors, for which the authors coined the term SMuRF (SMuRF = hypercholesterolemia, hypertension, diabetes, smoking) [1]. Their single-center study in Sydney, Australia, found that the prevalence of SMuRF-less STEMI patients had increased from 11% in 2006 to 27% in 2014. However, SMuRF status was not associated with extent of coronary disease or in-hospital outcomes [1]. This report was followed in 2019 by an expanded study from 2 Australian registries (GRACE, CONCORDANCE), which included 42 hospitals, and which confirmed the increasing SMuRF-less prevalence in STEMI presentations over time (to 24%). However, that study now reported a higher in-hospital mortality rate in SMuRF-less patients [2]. The need to assess longer-term outcomes was stressed [2]. 

An earlier 2015 report from the Canadian GRACE registry, while not using the SMuRF acronym, evaluated the prevalence and prognosis of over 3,800 STEMI patients with 0, 1–2, or 3–4 traditional risk factors, and who presented between 1999–2008 [3]. Traditional risk factors were absent in 14.5%, and these patients were at increased risk of in-hospital mortality. 

A most recent study used the SWEDEHEART registry database to assess SMuRF-less STEMI prevalence and outcomes [4]. Among 62,048 patients with STEMI registered between 2005 and 2018, 14.9% had no SMuRF. These patients were at increased risk for in-hospital mortality, which was particularly evident in women [4]. However, early mortality rates were attenuated after adjustment for the use of guideline-indicated treatments during the immediate post-infarction period. 

The foregoing reports originated from outside the United States, where potential differences in demographics and management practices could have influenced the generalizability of results. However, a 2011 report by Canto et-al. used the US National Registry of Myocardial Infarction (NRMI) for 1994–2006 and addressed the association of the number of coronary risk factors and mortality after a first myocardial infarction [5]. They observed a 14.4% prevalence of patients with no standard risk factors and reported an inverse association between hospital mortality and the number of risk factors. 

In summary, these reports, covering the past 2 decades and more, support an increasing proportion of SMuRF-less STEMI presentations over time. However, they are inconsistent and inconclusive with respect to the question of the relative risk of in-hospital mortality, and they provide limited information on long-term outcomes. Our findings update and complement these prior observations: the proportion of SMuRF-less STEMI patients indeed has grown and now is substantial (at least one-quarter of STEMI patients); further, in the modern era of more aggressive in-hospital and subsequent MI care, the mortality gap observed in some earlier studies appears to have closed. 

**Mechanistic Considerations and Non-Standard Risk Factors**: The observation of a substantial proportion of SMuRF-less STEMI patients points to the importance of non-standard risk factors. The classic international INTERHEART case-control study of myocardial infarction reported that, at a population level, 90–94% of population attributable risk could be accounted for by 9 modifiable risk factors. To the standard 4 factors, INTERHEART added dietary patterns, physical activity, alcohol consumption, waist/hip ratio, and psychosocial factors to achieve this high level of risk prediction [6]. An important developing environmental risk factor is air pollution [7].

Additional proposed non-traditional risk factors fall into 3 general categories: biomarkers, genetics, and imaging tests. Of new lipid biomarkers, lipoprotein(a) and apolipoprotein-B (ApoB) show promise for improving risk stratification and as targets for treatment [8,9,10]. Other targets for treatment include PCKS9i (current), AngPTL3, and apoCIII (future). 

The MORGAM project tested 30 other novel biomarkers from different pathophysiological pathways in 7915 patients with 538 incident 10-year cardiovascular events, from which a biomarker score was developed and validated in a separate population. The addition of a biomarker score that included *n*-terminal pro-brain natriuretic peptide, C-reactive protein, and a sensitive troponin-I assay to a conventional risk model improved 10-year MACE estimates [11]. In contrast, an earlier report from the Atherosclerosis Risk in Communities) (ARIC) study did not find incremental value in adding any of 19 novel risk markers to existing coronary heart disease risk models [12]. 

With the current facility in performing individual whole exome and whole genome analyses, investigators are exploring monogenetic and polygenic risk scores as an approach to improving the prediction of atherosclerotic cardiovascular risk. In a US whole-genome study of 2081 patients from 4 racial groups hospitalized with early-onset MI, both familial hypercholesterolemia mutations and a high polygenic score were associated with a >3-fold increased odds of early-onset MI [13]. However, the high polygenic score had a 10-fold greater prevalence than the monogenic score. 

The UK Biobank Cardio-Metabolic Consortium CHD Working Group used a meta-analytic approach to combine large-scale (22,242 CAD cases, 460,387 non-cases), genome-wide, and targeted genetic association data to develop a genomic risk score for CAD (metaGRS), which consisted of 1.6 million genetic variants [14]. Those in the top 20% of metaGRS had a hazard ratio of 4.17 compared to the bottom 20%. The metaGRS had a higher C-index for incident CAD than any of 6 conventional risk factors. Fortunately, an aggressive approach to standard risk factor reduction appears to be particularly impactful in those with mono- or high polygenetic risk [15].

Assigning ASCVD risk by risk factor, biomarker, and genetic testing is inherently probabilistic, whereas imaging can provide direct evidence of atherosclerosis or its absence in individual patients. Of non-invasive imaging tests applicable for wide-spread testing for CAD, coronary artery calcium (CAC) scoring using computed tomography has emerged as the leading contender, and its expanding use for CAD risk refinement in not-low risk patients is advocated by current prevention guidelines [16,17,18,19]. Silverman et-al explored the impact of CAC on the risk of CAD events in individuals at the extremes of traditional risk factor burden in the Multi-Ethnic Study of Athersclerosis (MESA). They found that at these extremes of risk factor burden, CAC distribution was heterogeneous and that a high CAC burden even among individuals without risk factors was associated with an elevated event rate. With this in mind, our investigative group performed a randomized vanguard study (CorCal) comparing statin selection by CAC score versus the traditional pooled cohort equation in 601 patients at primary ASCVD risk [20]. CAC-guidance was found to be a more efficient, cost-effective, motivating approach with a low event rate. A larger comparative outcomes trial is now underway (CorCal Outcomes). 

Clinical Implications: Our study together with others in the past 5 years emphasize the importance of recognizing patients at risk for and presenting with STEMI but without the 4 major traditional modifiable risk factors. Additional risk factors now may need to be considered in assessing primary coronary risk, including other lifestyle, environmental, and lipid markers, e.g., Lp (a) and ApoB, as well as imaging scores (i.e., CAC), with genetic testing reserved for suspected monogenetic disease and, in the near future, for polygenetic risk. Given earlier reports of poor in-hospital outcomes, SMuRF-less patients presenting with STEMI should not be considered inherently at lower risk, but they should be managed as aggressively with contemporary in-hospital and outpatient therapies as those with traditional risk factors. 

Advanced imaging techniques, such as optical coherence tomography (OCT) and, to a lesser extent, intravascular ultrasonography (IVUS), can be helpful in this group of SMuRF-less STEMI patients in defining the characteristics and, hence, pathophysiology of culprit (and non-culprit) lesions in SMuRF-less STEMI patients [21].

In addition, novel cardiovascular preventive therapies are emerging, which may be relevant to SMuRF-less patients at risk of a first or recurrent STEMI. Increased circulating and myocardial concentrations of inflammatory cytokines such as interleukins -1 and -6 represent important metabolic factors predisposing to cardiovascular disease, and inhibition of these cytokines may reduce risk [22]. Similarly, the SGLT2 inhibitors (gliflozins) are emerging as major cardiovascular preventive therapies, not only in diabetics, but increasingly in non-diabetics as well [23].

**Strengths and Limitations**: A strength of the study is its use of a dedicated, prospectively collected catheterization lab database integrated with a long-standing institutional electronic medical records system. Another strength is a long-standing, efficient and system-wide approach to early reperfusion therapy of STEMI. A limitation is that of all retrospective, observational studies, in that results are subject to uncorrected selection bias, which, despite attempts at adjustment, may leave some selection bias unaccounted for. However, the consistency of our findings through multiple subgroup and sensitivity analyses has reassured us with respect to any important contribution of residual selection bias. Additionally, our population is primarily of European-American ancestry, so that results may not be generalizable to other ethnic/racial groups. Results also may not apply to healthcare systems with substantially different approaches to STEMI management. Another limitation is that we do not have information on prehospital STEMI deaths. Our comparison is therefore limited to outcomes occurring after initial hospital presentation. Some long-term events may have occurred outside of the Intermountain Healthcare system and be missed although we doubt that this would favor one over another group by SMuRF status.

## 5. Conclusions

In this large single-system US healthcare population, SMuRF-less STEMI presentation was confirmed to be common (26.2%). However, unlike some earlier, primarily non-US reports, early (<60 day) event rates were not higher in SMuRF-less patients, and long-term outcomes were favorable, with a reduced adjusted rate of total MACE and heart-failure hospitalization. Further studies to increase understanding, recognition, and treatment of non-standard risk factors in SMuRF-less subjects and to optimize STEMI management are indicated.

## Figures and Tables

**Figure 1 jcm-12-00075-f001:**
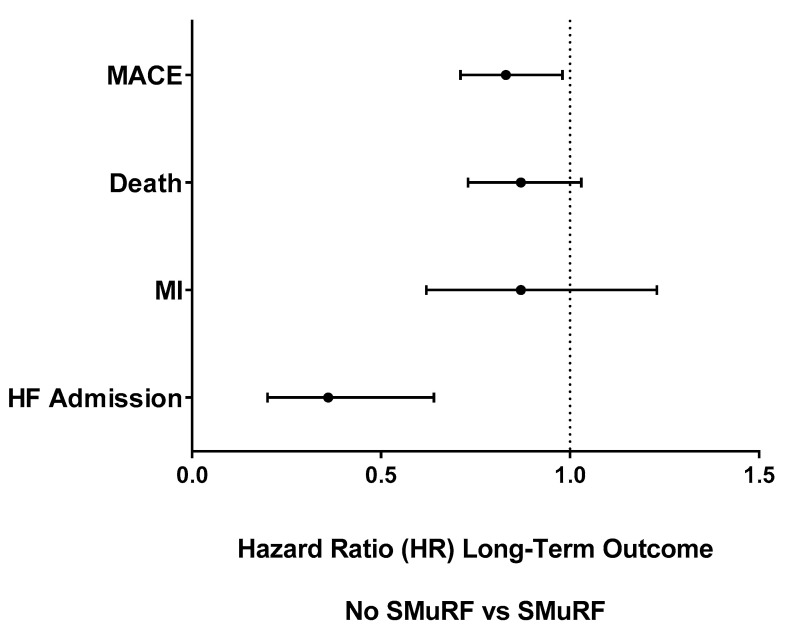
Long-term Outcomes Hazard Ratios (HRs) in SMuRF-less vs. SMuRF (referent) STEMI Patients.

**Figure 2 jcm-12-00075-f002:**
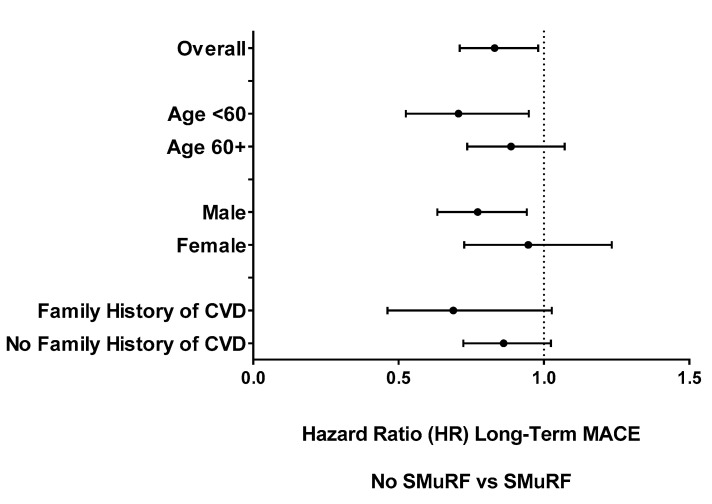
Long-term MACE Hazard Ratios (HR) for Subgroups by SMuRF Status.

**Table 1 jcm-12-00075-t001:** Baseline Characteristics of STEMI Patients by SMuRF Status.

	SMuRF		No SMuRF		
Demographics and Clinical Characteristics	*n* = 2591		*n* = 919		*p*-Value
	*n*	%	*n*	%	
**Age, median (IQR)**	61 (53, 71)		61 (52, 70)		0.09
**Age groups**					0.02
<40	85	3.3%	49	5.3%	
40–49	360	13.9%	140	15.2%	
50–59	720	27.8%	228	24.8%	
60–69	717	27.7%	271	29.5%	
70–79	471	18.2%	150	16.3%	
>79	238	9.2%	80	8.7%	
**Sex**					0.009
Male	1885	72.8%	709	77.3%	
Female	706	27.3%	210	22.9%	
**Race**					0.28
White/Caucasian	2260	87.2%	818	89.0%	
African American	14	0.5%	8	0.9%	
Asian	57	2.2%	15	1.6%	
Pacific Islander	5	0.3%	3	0.3%	
Unknown	255	9.8%	75	8.2%	
**Family history of heart disease**	849	32.8%	137	14.9%	<0.0001
**Comorbidities**					
Atrial Fibrillation (AF)	420	16.2%	112	12.2%	0.004
COPD	234	9.0%	52	5.7%	0.001
Depression	438	16.9%	136	14.8%	0.14
Heart Failure (HF)	126	4.9%	33	3.6%	0.11
Stroke	37	1.4%	12	1.3%	0.7
**SMURF criteria**					
Diabetes	1044	40.3%	0		NA
Hyperlipidemia	1613	62.3%	0		NA
Hypertension	1736	67.0%	0		NA
Smoking history					NA
Never	1785	68.9%	919		
Former	285	11.0%	0		
Current	521	20.1%	0		

IQR = Interquartile Range. COPD = chronic obstructive pulmonary disease. Analyses: Chi-square (categorical) and Wilcoxon rank sum (continuous) tests were used to examine differences in baseline characteristics for those patients with and without SMuRF.

**Table 2 jcm-12-00075-t002:** Interventions and Medications for STEMI Patients by SMuRF Status.

	SMuRF		No SMuRF		
Treatments and Medications	*n* = 2591		*n* = 919		
	*n*	%	*n*	%	*p*-Value
PCI performed	2267	87.5%	788	85.8%	0.17
CABG	210	8.1%	58	6.3%	0.08
**Discharge Medications**					
Beta Blocker	2177	84.0%	787	85.6%	0.25
ACE-I/ARB	1739	67.1%	576	62.7%	0.01
Anticoagulant	887	34.2%	333	36.2%	0.27
Antiplatelet	2464	95.1%	887	96.5%	0.50
Aspirin	2454	94.7%	875	95.2%	0.65
CCB	314	12.1%	100	10.9%	0.32

PCI = percutaneous coronary intervention; CABG = coronary artery bypass graft surgery; ACE-I = angiotensin converting enzyme inhibitor; ARB = angiotensin receptor blocker; CCB = calcium channel blocker. Analyses: Chi-square tests were used to examine differences in treatments and medications for those patients with and without SMuRF.

**Table 3 jcm-12-00075-t003:** Outcomes of STEMI Patients by SMuRF Status.

	SMuRF		No SMuRF					
	*n* = 2591		*n* = 919					
	*n*	%	*n*	%	Unadjusted *p*-Values	Adj ^a^ HR	95% CI	*p*-Value
**60-day Outcomes**								
MACE	210	8.1%	71	7.7%	0.76	0.95	(0.72, 1.25)	0.72
Death	170	6.6%	65	7.1%	0.57	1.06	(0.79, 1.42)	0.69
MI	14	0.5%	4	0.4%	0.70	NA ^b^		
HF Hospitalization	27	1.0%	2	0.2%	0.03	NA ^b^		
**Long-term Outcomes**								
MACE	813	31.4%	197	21.4%	0.02	0.83	(0.71, 0.98)	0.02
Death	646	24.9%	160	17.4%	0.11	0.87	(0.73, 1.03)	0.11
MI	178	6.9%	42	4.6%	0.29	0.87	(0.62, 1.23)	0.44
HF Hospitalization	125	4.8%	13	1.4%	0.0003	0.36	(0.20, 0.64)	0.0005

Analysis: Cox proportional hazard regression was used to examine outcomes adjusted for baseline differences comparing No SMuRF vs. SMuRF. ^a^ Adj = No SMuRF vs. SMuRF adjusted for age, sex, ACE-I/ARB, AF, COPD, family history. ^b^ NA = no modeling done due to too few outcomes. ACE-I = angiotensin converting enzyme inhibitors; AF = atrial fibrillation; COPD = chronic obstructive pulmonary disease; HF = heart failure; MACE = major adverse cardiovascular event; MI = myocardial infarction; HR = hazard ratio.

**Table 4 jcm-12-00075-t004:** Outcomes of STEMI Patients by SMuRF Count.

	SMuRF Count													
	0		1		2		3		4					
	*n* = 919		*n* = 834		*n* = 1007		*n* = 649		*n* = 101					
	*n*	%	*n*	%	*n*	%	*n*	%	*n*	%	Unadjusted	Adj ** HR	95% CI	*p*-Value
											*p*-Values			
**60-day Outcomes**														
MACE	71	7.7%	61	7.3%	78	7.8%	65	10.0%	6	5.9%	0.30	1.07	(0.96, 1.19)	0.23
Death	65	7.1%	50	6.0%	64	6.4%	52	8.0%	4	4.0%	0.92	1.03	(0.91, 1.15)	0.68
MI	4	0.4%	3	0.4%	8	0.8%	2	0.3%	1	1.0%	0.63	NA *		
HF Hospitalization	2	0.2%	8	0.2%	7	0.7%	11	1.7%	1	1.0%	0.01	NA *		
**Long-term Outcomes**														
MACE	197	21.4%	244	29.3%	292	29.0%	228	35.1%	49	48.5%	<0.0001	1.16	(1.09, 1.23)	<0.0001
Death	160	17.4%	193	23.1%	229	22.7%	184	28.4%	40	39.6%	<0.0001	1.17	(1.09, 1.25)	<0.0001
MI	42	4.6%	62	7.4%	59	5.9%	46	7.1%	11	10.9%	0.08	1.11	(0.98, 1.25)	0.11
HF Hospitalization	13	1.4%	34	4.1%	44	4.4%	38	5.9%	9	8.9%	<0.0001	1.41	(1.21, 1.64)	<0.0001

Analysis: Cox proportional hazard regression was used to examine outcomes adjusted for baseline differences comparing each one count increase in SMuRF. Adj ** = Each one count increase in SMuRF adjusted for age, sex, ACE-I / ARB, CCB, AF, COPD, depression, HF, family history. * NA = no modeling done due to too few outcomes. ACE-I = angiotensin converting enzyme inhibitors; AF = atrial fibrillation; COPD = chronic obstructive pulmonary disease; HF = heart failure; MACE = major adverse cardiovascular event; MI = myocardial infarction; HR = hazard ratio.

## Data Availability

The data underlying this article cannot be shared publicly due to US privacy laws. Data are available upon reasonable request to the corresponding author.

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
