# Peer review of "Cardiovascular Outcomes of ST-Elevation Myocardial Infarction (STEMI) Patients without Standard Modifiable Risk Factors (SMuRF-Less): The Intermountain Healthcare Experience"

_jcm, 2022, doi:10.3390/jcm12010075_

Round 1
Reviewer 1 Report
Manuscript titled " Cardiovascular Outcomes of ST-Elevation Myocardial Infarction (STEMI)
Patients without Standard Modifiable Risk Factors (SMuRF-less): The Intermountain Healthcare Experience" is a very interessing original article about the link between SMuRF and STEMI incidence . The overall structure is of good quality, figures are appropriate to this journal and clear to readers. References are of good quality and updated in this field. However the authors should improve the manuscript in several parts, such as:
1 In introduction or discussion, authors should highlight on metabolic risk factors associated also to high risk of HF and other CVD ( i.e, high levels of plasma Interleukin 1; please cite doi: 10.26355/eurrev_202111_27124. )
2 In discussion, a proper description also of the cardioprotective effects of glifozins. For example empaglifozin that could reduce cardiovascular risk factors (you can cite 10.1186/s12933-021-01346-y).
Author Response
Dear JCM,
We thank the editor for the opportunity to respond to the reviewers’ comments and requests and to submit a revised manuscript for consideration for publication in J Clin Med. See below for a point-to-point response. We hope that the revised MS now will be acceptable for publication in JCM.
Sincerely yours,
Jeffrey L Anderson
Corresponding author.
Reviewer 1
…However, the authors should improve the manuscript in several parts, such as:
1 In introduction or discussion, authors should highlight on metabolic risk factors associated also to high risk of HF and other CVD ( i.e, high levels of plasma Interleukin 1; please cite doi: 10.26355/eurrev_202111_27124. )
2 In discussion, a proper description also of the cardioprotective effects of glifozins. For example empaglifozin that could reduce cardiovascular risk factors (you can cite 10.1186/s12933-021-01346-y).
Response: We thank the reviewer for the positive comments about our MS. We also appreciate the suggested additional references with their important implications. We now have added both of them to the discussion in the section on “Mechanistic Considerations…” as references 22 and 23. We also have added the following accompanying text: “Novel cardiovascular preventive therapies are emerging, which may be relevant to SMuRF-less patients at risk of a first or recurrent STEMI. Increased circulating and myocardial concentrations of inflammatory cytokines such as interleukins -1 and -6 represent important metabolic factors predisposing to cardiovascular disease, and inhibition of these cytokines may reduce risk(22). Similarly, the SGLT2 inhibitors (gliflozins) are emerging as major CV preventive therapies, not only in diabetics, but increasingly in non-diabetics as well(23).
Reviewer 2 Report
I congratulate the authors for their research concerning the prevalence and outcomes of the patients with STEMI, but without modifiable risk factors.
I suggest adding to the mechanistic considerations some data provided by intracoronary imaging (mainly Optical Coherence Tomography -OCT but also, to a lesser extent IVUS) regarding STEMI non-atherosclerotic culprit lesions, as spontaneous coronary dissections, intramural haematoma or honney-comb like lesions (please refer to 10.3390/jcm11010265). These kind of lesions are not predicted by the currently accepted coronary risk factors.
Author Response
Dear JCM,
We thank the editor for the opportunity to respond to the reviewers’ comments and requests and to submit a revised manuscript for consideration for publication in J Clin Med. See below for a point-to-point response. We hope that the revised MS now will be acceptable for publication in JCM.
Sincerely yours,
Jeffrey L Anderson
Reviewer 2
I congratulate the authors for their research concerning the prevalence and outcomes of the patients with STEMI, but without modifiable risk factors.
I suggest adding to the mechanistic considerations some data provided by intracoronary imaging (mainly Optical Coherence Tomography -OCT but also, to a lesser extent IVUS) regarding STEMI non-atherosclerotic culprit lesions, as spontaneous coronary dissections, intramural haematoma or honney-comb like lesions (please refer to 10.3390/jcm11010265). These kind of lesions are not predicted by the currently accepted coronary risk factors.
Response: We thank the reviewer for the positive comments about our MS. We also appreciate the suggested additional reference with its important implications. We now have added this reference to the discussion in the section on “Mechanistic Considerations…” as reference 21. We also have added the following accompanying text. “Advanced imaging techniques, such as optical coherence tomography (OCT) and, to a lesser extent, intravascular ultrasonography (IVUS), can be helpful in this group of SMuRF-less STEMI patients in defining the characteristics and, hence, pathophysiology, of culprit (and non-culprit) lesions in SMuRF-less STEMI patients(21).”